# Digital Domain TDI-CMOS Imaging Based on Minimum Search Domain Alignment

**DOI:** 10.3390/s25113490

**Published:** 2025-05-31

**Authors:** Han Liu, Shuping Tao, Qinping Feng, Zongxuan Li

**Affiliations:** 1Changchun Institute of Optics, Fine Mechanics and Physics, Chinese Academy of Sciences, Changchun 130033, China; liuhan22@mails.ucas.ac.cn (H.L.); fengqinping@ciomp.ac.cn (Q.F.); lizongxuan@ciomp.ac.cn (Z.L.); 2University of Chinese Academy of Sciences, Beijing 100049, China

**Keywords:** time delay integration, image alignment, image motion computational model, subpixel

## Abstract

In this study, we propose a digital domain TDI-CMOS dynamic imaging method based on minimum search domain alignment, which consists of five steps: image-motion vector computation, image jitter estimation, feature pair matching, global displacement estimation, and TDI accumulation. To solve the challenge of matching feature point pairs in dark and low-contrast images, our method first optimizes the size and position of the search box using an image motion compensation mathematical model and a satellite platform jitter model. Then, the feature point pairs that best match the extracted feature points of the reference frame are identified within the search box of the target frame. After that, a kernel density estimation algorithm is proposed for calculating the displacement probability density of each feature point pair to fit the actual displacement between two frames. Finally, we align and superimpose all the frames in the digital domain to generate a delayed integral image. Experimental results show that this method greatly improves the alignment speed and accuracy of dark and low-contrast images during dynamic imaging. It effectively mitigates the effects of image motion and jitter from the spatial camera, and the fitted global image motion error is kept below 0.01 pixels, which is compensated to improve the MTF coefficient of the image motion and jitter link to 0.68, thus improving the imaging quality of TDI.

## 1. Introduction

High-resolution optical remote sensing satellites hold significant value in various fields, including Earth observation, military reconnaissance, geographic mapping, and resource surveys. Complementary metal–oxide–semiconductor (CMOS) devices equipped with time delay integration (TDI) functionality are widely utilized in high-resolution satellite image acquisition due to their high signal-to-noise ratio (SNR) and sensitivity. However, the imaging quality of TDI is influenced by several factors, such as image motion and satellite platform jitter [1].

Image motion occurs during normal push-broom scanning, where the image of the ground target and the image detector surface experience relative motion. Orbital parameters and satellite attitude typically determine the direction and magnitude of this motion. In dynamic scenes, rapid movement of the target can lead to blurring and displacement of the image during exposure, which in turn affects the clarity and accuracy of the final image [2,3].

Current methods for compensating image motion can be categorized into four types [4,5,6]: mechanical, optical, electronic, and image-based techniques. The image-motion compensation of mainstream high-resolution remote sensing satellites mainly relies on a combination of mechanical adjustment of the bias angle and electronic correction of the magnitude of the image-motion velocity. Specifically, the bias mechanism is calibrated based on the calculated results of the image-motion vector to correct for angular deviations, while the magnitude of motion speed deviation is addressed by adjusting the transmission frequency of the TDI line.

However, the incorporation of additional mechanical structures introduces significant complexity, increasing the size and mass of the payload and posing challenges in meeting the high-speed, multi-directional, and high-frequency imaging requirements of agile satellites. In light of these limitations, advancements in onboard computational capabilities and digital image processing technologies have led to a growing emphasis on image-based self-compensation for image motion. This innovative approach has garnered considerable attention and is rapidly evolving, presenting a promising alternative to traditional compensation methods.

Satellite platform jitter also significantly affects TDI imaging quality [7]. Satellite jitter arises from attitude adjustments and the periodic motion of satellite components during orbital operations. Due to the high sensitivity and uncontrollability of satellite jitter, its effects are challenging to mitigate through physical means [8]. Offsets occur when satellite jitter is transmitted to the image plane, altering the geometric and radiometric information of the image and directly leading to a degradation in image quality. The primary jitter sources on the satellite platform include the solar sail panel, the magnetic torque rod, the attitude control flywheel, and the control moment gyroscope (CMG). These jitter sources cause distortion and blurring in TDI images, and the impact of satellite platform jitters on imaging quality worsens as the spatial resolution of the payload increases [9].

For the compensation of satellite platform jitter effects, one is to use the data collected by the attitude sensor for direct image distortion correction, and the other is to reconstruct the degraded image to its original form by image restoration through Wiener filtering, constrained least squares filtering, inverse filtering, the Lucy–Richardson algorithm, and so on, using the platform tremor a priori knowledge [10].

However, these methods have notable shortcomings. First, resampling techniques may result in the loss of image information when addressing high-frequency jitter, particularly in the presence of multiple vibration sources on the starboard platform. Second, reconstruction methods that rely on a priori knowledge are highly dependent on the accuracy of the jitter model; inaccuracies in the model can significantly diminish reconstruction quality. Additionally, existing techniques often struggle to achieve optimal compensation effects in complex scenes with multi-source dithering, leading to unsatisfactory final image quality.

This paper introduces an innovative digital domain TDI-CMOS imaging technique that utilizes minimum search domain alignment to address these challenges. This approach aims to simultaneously mitigate the effects of satellite platform jitter on TDI imaging throughout the imaging process. The proposed method is specifically designed for the area-array push-scan imaging mode. It facilitates the integration of area-array dark and low-contrast images by optimizing the image alignment process and estimating the global displacement between frames. In the following sections, we will elaborate on the underlying principles of the method and evaluate its effectiveness in practical applications.

The remainder of this paper is organized as follows: Section 2 presents the working principle of the digital domain TDI-CMOS. Section 3 describes each imaging process of this method. Section 4 provides a simulation experiment of the digital domain TDI-CMOS imaging process. The simulation analysis is discussed in Section 5. Finally, the conclusion of this study is given.

## 2. Digital Image Domain TDI-CMOS Operating Principle

TDI-CMOS is a widely used imaging technology in remote sensing. Its working principle is based on TDI technology, wherein the image sensor comprises multiple rows, each capable of independently receiving optical signals. When the space camera and the scene move relative to each other, each row of the sensor captures optical signals from the same target at different times. This process enhances imaging quality and improves the SNR by successively sampling the same target across multiple time windows.

In contrast to traditional analog TDI, the digital domain TDI technique motions the charge signal integration process to an off-chip field-programmable gate array (FPGA) or processor, where the analog signal has been converted into digital format. In off-chip TDI imaging, the line time, frame period, null time, and number of TDI stages are calculated based on the requirements of TDI imaging. The CMOS then adjusts the drive timing according to these parameters, continuously outputting signals obtained from multiple exposures of the same target. These digital signals are synchronized with optical scanning and superimposed in the FPGA with memory, resulting in an integrated digital signal output [11].

The digital domain TDI is commonly employed in area-scanning imaging mode, where the area-scanning detector continuously captures images at a high frame rate along the track direction. This imaging mode allows for acquiring multi-frame images with high overlaps of the same scene. As shown in Figure 1, the imaging mode under which the satellite flies along its orbit and images a ground scene takes place at the sensor plane through the lens, and multiple frames of high overlap images $A_{1}$ to $A_{4}$ of the same scene A can be obtained, which are aligned to realize the superposition of the digital images in the off-chip FPGAs. The original frames, which may have low SNR, can be matched and superimposed, improving the SNR of the final output image by a factor of $\sqrt{m}$ on M-stage TDI. Additionally, redundant information in the highly overlapping multi-frame images provides significant flexibility for both spaceborne and ground processing.

Our previous work [12] proposed an off-chip digital domain TDI-CMOS camera and an image-motion self-compensation algorithm. In this study, we build upon this foundation by simultaneously implementing self-compensation for inter-frame image motion and tremor in the digital domain, without relying on a mechanical deflection angle adjustment mechanism.

## 3. TDI Imaging Method Based on Minimum Search Domain

This method comprises five steps: (1) calculating the image-motion vector at the moment of imaging, (2) estimating the image jitter volume, (3) matching feature point pairs within a minimum search domain, (4) estimating the global displacement volume, and (5) aligning and performing TDI accumulation. The specific flow is illustrated in Figure 2.

### 3.1. Image-Motion Vector Calculation

Currently, the primary method for calculating image displacement is based on the coordinate transformation of the image displacement velocity vector. This approach necessitates the use of matrix transformation to establish a precise coordinate relationship between the object point and the image point. Subsequently, the coordinate equation of the image point concerning the target’s image plane position is derived through coordinate correspondence. The image displacement velocity vector equations for the points on the image plane can be obtained by differentiating this equation concerning time.

As shown in Figure 3, according to the 11 matrix linear transformations of 7 coordinate systems proposed in the literature [13] as an example, it has gone through the geographic coordinate system G($G_{1}$, $G_{2}$, $G_{3}$) to the earth coordinate system E($E_{1}$, $E_{2}$, $E_{3}$) to the geocentric inertial coordinate system I($I_{1}$, $I_{2}$, $I_{3}$) to the spacecraft orbit coordinate system B($B_{1}$, $B_{2}$, $B_{3}$) to the spacecraft coordinate system S($S_{1}$, $S_{2}$, $S_{3}$) to the camera coordinate system C($C_{1}$, $C_{2}$, $C_{3}$) and finally to the image plane coordinate system P($P_{1}$, $P_{2}$, $P_{3}$) of the seven transformations. The scene in the image plane at the coordinates of P($P_{1}$, $P_{2}$) can be calculated by its coordinates in the geographical coordinate system G($G_{1}$, $G_{2}$). The process for calculating the image motion velocity vector on the image plane is as follows:(1)$$P=\left[\begin{array}{c} P_{1}\\ P_{2}\\ P_{3}\\ 1 \end{array}\right]\;=\left[\begin{array}{cccc} -\frac{f}{H-h} & 0 & 0 & 0\\ 0 & -\frac{f}{H-h} & 0 & 0\\ 0 & 0 & \frac{f}{H-h} & -f\\ 0 & 0 & 0 & 1 \end{array}\right]\left[\begin{array}{cccc} 1 & 0 & 0 & 0\\ 0 & cos\varphi & sin\varphi & 0\\ 0 & -sin\varphi & cos\varphi & 0\\ 0 & 0 & 0 & 1 \end{array}\right]\left[\begin{array}{cccc} cos\theta & 0 & -sin\theta & 0\\ 0 & 1 & 0 & 0\\ sin\theta & 0 & cos\theta & 0\\ 0 & 0 & 0 & 1 \end{array}\right]\times \;\left[\begin{array}{cccc} cos\psi & sin\psi & 0 & 0\\ -sin\psi & cos\psi & 0 & 0\\ 0 & 0 & 1 & 0\\ 0 & 0 & 0 & 1 \end{array}\right]\left[\begin{array}{cccc} 1 & 0 & 0 & 0\\ 0 & 1 & 0 & 0\\ 0 & 0 & 1 & -\left(R+H\right)\\ 0 & 0 & 0 & 1 \end{array}\right]\left[\begin{array}{cccc} cos\gamma & 0 & -sin\gamma & 0\\ 0 & 1 & 0 & 0\\ sin\gamma & 0 & cos\gamma & 0\\ 0 & 0 & 0 & 1 \end{array}\right]\times \;\left[\begin{array}{cccc} cosi_{0} & -sini_{0} & 0 & 0\\ sini_{0} & cosi_{0} & 0 & 0\\ 0 & 0 & 1 & 0\\ 0 & 0 & 0 & 1 \end{array}\right]\left[\begin{array}{cccc} cos\omega t & 0 & sin\omega t & 0\\ 0 & 1 & 0 & 0\\ -sin\omega t & 0 & cos\omega t & 0\\ 0 & 0 & 0 & 1 \end{array}\right]\left[\begin{array}{cccc} cosi_{0} & sini_{0} & 0 & 0\\ -sini_{0} & cosi_{0} & 0 & 0\\ 0 & 0 & 1 & 0\\ 0 & 0 & 0 & 1 \end{array}\right]\times \;\left[\begin{array}{cccc} cos\gamma_{0} & 0 & sin\gamma_{0} & 0\\ 0 & 0 & 0 & 0\\ -sin\gamma_{0} & 0 & cos\gamma_{0} & 0\\ 0 & 0 & 0 & 1 \end{array}\right]\left[\begin{array}{cccc} 1 & 0 & 0 & 0\\ 0 & 1 & 0 & 0\\ 0 & 0 & 1 & \left(R+h\right)\\ 0 & 0 & 0 & 1 \end{array}\right]\left[\begin{array}{c} G_{1}\\ G_{2}\\ 0\\ 1 \end{array}\right]$$

Differentiating Equation (1) with respect to time t yields the equation for the image motion velocity at the image plane:(2)$${\frac{dP}{dt}|}_{t=0}=\left[\begin{array}{c} dP_{1}/dt\\ dP_{2}/dt\\ dP_{3}/dt\\ 0 \end{array}\right]=\left[\begin{array}{c} V_{p_{1}}\\ V_{p_{2}}\\ V_{p_{3}}\\ 0 \end{array}\right]$$

In this context, R represents the radius of the Earth relative to its center; H is the spacecraft orbital altitude; f is the focal length of the camera lens; h is the height of the feature terrain at the photographed target; φ, θ, and ψ are the roll, pitch, and yaw attitude angles of the spacecraft’s coordinate system for the orbital coordinate system, respectively; $\gamma =\gamma_{0}+\Omega t$, with $\gamma_{0}$ being the central angle corresponding to the point of descending crossover of the spacecraft to the orbital plane at the moment of photo-graphing, and Ω being the average angular rate of spacecraft rotation around the earth’s center along the orbit; $i_{0}$ is the orbital inclination; and ω is the angular rate of earth rotation.

The calculated values $V_{p1}$, $V_{p2}$ represent the image motion velocities in the X and Y direction of the image surface. For a single frame exposure time $T_{stage}$, the image motion during the imaging time interval from the 1st frame to the *k*th frame, can be expressed in terms of the deviation of the image motion in the rows and columns of the image:(3)$$∆x_{motion}=T_{stage}\times \sum_{i=1}^{k-1}{i\times V}_{p_{1}}\left(i\right)$$(4)$$∆y_{motion}=T_{stage}\times \sum_{i=1}^{k-1}{i\times V}_{p_{2}}\left(i\right)$$

Similarly, as shown in Figure 4, for an image detector with pixel size a, the number of pixels motioned in the row and column directions of the image in the imaging plane during the time interval $(k-1)T_{stage}$ between the 1st frame and the *k*th frame can be expressed as follows:(5)$$∆m_{motion}=\frac{T_{stage}\times \sum_{i=1}^{k-1}{i\times V}_{p_{1}}\left(i\right)}{a}$$(6)$$∆n_{motion}=\frac{T_{stage}\times \sum_{i=1}^{k-1}{i\times V}_{p_{2}}\left(i\right)}{a}$$

### 3.2. Image Jitter Estimation

During Earth observation using space-based optical remote sensors, platform jitters are characterized by small amplitudes, a wide frequency spectrum, and various forms. The amplitude typically occurs on the sub-pixel scale, ranging from 0.1 Hz to 1 kHz [10]. These jitters are often not simply harmonic or linear; instead, they manifest as the superposition of multiple jitter sources and combinations of motions. This complexity complicates predicting and inhibiting jitter amplitudes at the imaging focal plane at any given moment.

In this paper, we examine combined jitters as a case study to investigate the displacement of a focal plane image resulting from the superposition of low-frequency, high-frequency, and random jitter sources [14]. Their functional distributions and key parameters are presented in Table 1.

According to the 3σ quasi-measurement, for a single random jitter that follows a normal distribution X~N(μ,σ), there is 99.73% certainty that its amplitude is within the interval (μ−3σ,μ+3σ). The maximum amplitude of this single random jitter can be expressed as follows:(7)$$A_{random}=\left|\mu \right|+3\;\sigma$$

The maximum amplitude of the mixed jitter resulting from the superposition of multiple low-frequency, high-frequency, and random jitter sources can be expressed as follows:(8)$$A_{max}=\sum A_{Low\text{-}frequency}+A_{High\text{-}frequencyy}+A_{random}$$

### 3.3. Feature Pairs Matching

Traditional image-matching methods typically employ the same feature description operator to extract all feature points from two frames. During the subsequent matching point pair search process, each feature point in the target frame is calculated and compared with every feature point in the reference frame to obtain the globally optimal matching feature point pair. However, when dealing with a large number of feature points, the matching point pair search can consume up to 70% of the computation time [15]. Additionally, due to the large frame width, low brightness, and low signal-to-noise ratio of single-frame surface-array TDI images, traditional image-matching methods are often computationally intensive and prone to matching errors, making it challenging to meet the high-precision and fast-response imaging requirements of on-planet TDI.

The feature point pair matching in the minimum search domain proposed in this paper consists of the following steps outlined in Figure 5:

First, all feature points in the reference frame and the frame to be matched are extracted using a null domain feature point operator. The SIFT (scale-invariant feature transform) operator is a classical and most commonly used feature extraction operator and is selected for this experiment due to its invariance to rotation, scale, and affine transformations, as well as for its stability to noise, viewing angle, and illumination [16].

In the second step, as shown in Figure 6, the position of the search box is determined based on the inter-frame image motion derived from the image motion vector mathematical model in Section 3.1, and the size of the search box is determined based on the maximum magnitude derived from the image motion modelling in Section 3.2; each feature point extracted from the reference frame is found to have a desired position in the target frame, and a search box is built with the center of the point.

Finally, the similarity measure, such as the Euclidean distance, is calculated between all feature points in the search box and their corresponding feature points in the base frame, leading to the identification of the matching feature point pair with the smallest distance.

Through the aforementioned three steps, all feature points in the base frame have identified matched candidates in the target frame. We then combine the matched point pairs into a set $Z=\{x_{n},y_{n}{\}}_{n=1}^{M}$. For any pair of matched points between the base frame and the target frame with coordinates $\left(x_{1i},y_{1i}\right)$ and $\left(x_{2i},y_{2i}\right)$, we can calculate their coordinate differences as follows:(9)$$\Delta x=x_{2i}-x_{1i}$$(10)$$\Delta y=y_{2i}-y_{1i}$$

The coordinate differences Δx and Δy of all feature points are then plotted as a scatter plot, which visually represents the displacement distribution between the feature points. By examining the shape and distribution of the scatter plot, we can initially assess the direction and magnitude of the global displacement.

### 3.4. Global Displacement Estimate

Kernel density estimation (KDE) is a non-parametric method that smooths data by placing a kernel function around each data point, thereby filling in the gaps between isolated observations and enabling the evaluation of the probability density of a random variable [17]. In Section 3.3, we obtained a set of matched feature point pairs. In this section, we will employ kernel density estimation to analyze the coordinate differences derived from these matched feature point pairs. These coordinate differences reflect the motion information between neighboring frames and can be utilized to estimate global displacement.

The formula for kernel density estimation is defined as follows [17]:(11)pn^x=1nr∑i=1nKxi−xn
where $x_{i}$ is any variable belonging to the overall data, *n* is the number of data points in the sample, *K* is the kernel density function, and r is the smoothing bandwidth that controls the degree of smoothing (with larger values of r resulting in smoother data). In this study, according to the literature [18], the mean integrated squared error (MISE) is used to measure the difference between the estimated density function and the true density function, and its minimum value is used to determine the optimal bandwidth r. And $x_{i}$ is a data point in the sample [19]. In this paper, the Gaussian distribution probability density function is employed as the kernel function, expressed as follows:(12)$$f\left(x\right)=\frac{1}{\sqrt{2\pi }}e^{-\frac{x^{2}}{2}}$$

Substituting Equation (12) into Equation (11) yields the Gaussian kernel density function as follows:(13)$$\overset{\wedge }{p_{n}}(x)=\frac{1}{nr}\frac{1}{\sqrt{2\pi }}\sum_{i=1}^{n}e^{-\frac{{\left(x_{i}-x\right)}^{2}}{2r^{2}}}$$

By analyzing the kernel density estimation results for each coordinate difference point, the weighted average of these coordinate differences serves as an estimate of the global displacement. The pixel displacements corresponding to the image motion deviation between the frame to be aligned and the reference frame in the row and column directions are denoted by Δm and Δn:(14)$$\Delta m=\frac{\sum_{n}\Delta x\cdot \overset{\wedge }{p_{n}}(\Delta x,\Delta y)}{n}$$(15)Δn=∑nΔy·pn^(Δx,Δy)n

### 3.5. TDI Accumulation

To align the images of each frame in the TDI accumulation process, we use M-stage integration as an example. The first frame in an imaging cycle serves as the reference frame, while the remaining M-1 frames are designated as the frames to be aligned. The integer parts of the pixel displacements in the row and column directions between the *k*th frame and the first frame are denoted as $\Delta m_{k}^{\prime }$ and $\Delta n_{k}^{\prime }$:(16)$$\Delta m_{k}^{\prime }=\left\lfloor \Delta m\right\rfloor$$(17)$$\Delta n_{k}^{\prime }=\left\lfloor \Delta n\right\rfloor$$

Respectively. The fractional parts of the pixel displacements $\Delta m_{k}^{\prime \prime }$ and $\Delta n_{k}^{\prime \prime }$ can be expressed as follows:(18)$$\Delta m_{k}^{\prime \prime }=\Delta m-\Delta m_{k}^{\prime }$$(19)$$\Delta n_{k}^{\prime \prime }=\Delta n-\Delta n_{k}^{\prime }$$

Since the gray value of an image pixel is proportional to the number of photogenerated charges, which correspond to the light-sensitive area and the luminance of the target object [12], the contribution of each pixel cell to the pixel gray value must be considered. The pixel array $P_{k}^{\prime }(x,y)$ of the *k*th frame image aligned with the first frame can be expressed as follows:(20)$$\begin{array}{ll} P_{k}^{\prime }(x,y)= & P(i+\Delta m_{k}^{\prime },j+\Delta n_{k}^{\prime })(1-\Delta m_{k}^{\prime \prime })(1-\Delta n_{k}^{\prime \prime })+\\ & P(i+\Delta m_{k}^{\prime },j+\Delta n_{k}^{\prime }+1)(1-\Delta m_{k}^{\prime \prime })\Delta n_{k}^{\prime \prime }+\\ & P(i+\Delta m_{k}^{\prime }+1,j+\Delta n_{k}^{\prime })\Delta m_{k}^{\prime \prime }(1-\Delta n_{k}^{\prime \prime })+\\ & P(i+\Delta m_{k}^{\prime }+1,j+\Delta n_{k}^{\prime }+1)\Delta m_{k}^{\prime \prime }\Delta n_{k}^{\prime \prime } \end{array}$$

Finally, by summing the inter-frame images of the digital domain TDI-CMOS camera after M-frame alignment, we can obtain the pixel gray value of each pixel cell of the M-stage TDI cumulative image $O_{M}$:(21)$$O_{M}(i,j)=P_{k}^{\prime }(i,j)+\sum_{l=1}^{M-1}P_{k-l}^{\prime }(i+\Delta m^{\prime }(k-l),j+\Delta n^{\prime }(k-l))$$

## 4. Experiments

### 4.1. Experimental Parameter Setting

To verify the effectiveness of the proposed TDI imaging technique, we selected a satellite remote sensing image from the Baotou range and the Songshan range shown in Figure 7a and Figure 7b, respectively, with sizes of 1024 pixel × 1024 pixel and 2048 pixel × 2048 pixel to simulate image detectors with different numbers of image elements. The satellite platform and camera parameters at their imaging moments are detailed in Table 2.

According to the image motion calculation model presented in Section 3.1, the image motion velocities in the along-track and across-track directions at the time of imaging are 0.01187 m/s and 0.001455 m/s, respectively. These velocities correspond to image motions of 0.9926 pixels and 0.1217 pixels between two adjacent frames in the along-track and across-track directions, respectively.

For the image jitter model, we established a hybrid jitter that encompasses low-frequency, high-frequency, and random jitters to simulate the jitter of the satellite platform during the imaging process. The parameters of this hybrid jitter are presented in Table 3, and the time-domain and frequency-domain spectra of the superimposed hybrid jitter are illustrated in Figure 8. The maximum amplitude $A_{max}$ of the hybrid jitter is 2 pixels, with a corresponding minimum search box size of 4 pixels by 4 pixels.

### 4.2. Analysis of Alignment Indicators

In this section, we analyze the effectiveness and advancements of the algorithm proposed in this paper regarding both speed and accuracy. Because the single-frame image is too dark and weak, it is necessary to lower the values of contrast-threshold, edge-threshold, Sigma, and other parameters in the SIFT operator as much as possible in order to extract as many feature points as possible to avoid incomplete extraction of feature points. So, we extracted varying numbers of feature points by applying different thresholds with the SIFT operator on the experimental images. We then matched the feature point pairs using both the SIFT global and minimum search domains within the same algorithmic framework. The time taken for the matching process is presented in Table 4, which indicates that our method significantly outperforms SIFT global pairing. Notably, as the search box size decreases, the time required for matching also decreases, particularly in scenarios with a large number of feature points, where the running speed can reach up to 3.86 times that of SIFT global pairing.

Figure 9 shows the comparison of the alignment effect of the dark and weak images between frames, and bolded parts indicate the shortest matching elapsed time for the same number of feature pairs. It can be seen that when the feature point extraction parameter threshold is set lower, a large number of mismatched points appear in the inter-frame displacement image, whereas the minimum search domain method aligns with fewer mismatched points and has better results in both experimental images.

The distribution of the differences between the coordinates of the feature point pairs in each frame and the first frame after pairing the images of the Baotou range is shown in Figure 10 and Figure 11. The horizontal and vertical axes represent the coordinate differences in the Y and X directions, respectively. The colors of the scattered points indicate the kernel density estimation of the differences, with denser distributions corresponding to higher probability densities. Compared to inter-frame matching using only SIFT, the scatter point distribution obtained through the minimum search domain method is more concentrated, effectively mitigating the influence of outliers on the results.

The coordinate differences of all feature point pairs in each frame are weighted and averaged to derive the global displacement estimates for each frame relative to the first frame. The error values are calculated by subtracting the global displacement estimates from the true displacements. As shown in Table 5, the minimum search domain method yields smaller error values compared to inter-frame matching using SIFT only, indicating greater accuracy in the inter-frame global displacement estimation. Figure 12 illustrates the distribution of error values from the minimum search domain algorithm across the 1st frame to the 96th frame. It is evident that the error values of the inter-frame displacements computed by this algorithm, along with the true displacements affected by image motion and platform tremor, remain within 0.01 pixels in both the across-track and along-track directions.

### 4.3. Analysis of TDI Imaging Results

After aligning each frame of the dark and weak image according to the estimated global image motion, TDI images were generated using integration stages 6, 12, 18, 24, 36, and 48. Additionally, TDI images were generated using an uncompensated method for comparison. The simulation results of this algorithm and the uncompensated method for stages 6 to 18 are presented in Figure 13, while the results for stages 24 to 48 are shown in Figure 14.

Table 6 displays the objective metrics of the images generated at each TDI stage, utilizing five metrics: structural similarity index (SSIM), peak signal-to-noise ratio (PSNR), cross-correlation (CC), and information entropy to evaluate the generated TDI images. SSIM evaluates the perceived similarity of two images through brightness, contrast, and structural information; the closer the value is to 1, the more similar the two images are. PSNR quantifies the fidelity of an image based on the pixel mean-square error; the higher the value is, the lower the distortion. CC measures the degree of linear correlation of the pixel values of the two images; the higher the value is, the better the global consistency. And entropy shows the complexity of image information, calculated by the probability of grayscale distribution; the richer the texture, the higher the entropy value. The four metrics are used to evaluate the image characteristics in terms of perceptual quality, numerical accuracy, statistical correlation, and informativeness dimensions, respectively [20], so we use these four metrics to evaluate the generated TDI images.

Observing Figure 13 and Figure 14, and Table 6, it can be found that the quality of the images generated by the algorithm is better than that of the uncompensated images at all stages, and the higher the stage, the higher the percentage of objective evaluation indexes improved.

MTF (modulation transfer function) is a comprehensive index describing the response of the whole imaging link to spatial frequency, reflecting the ability of the system to transfer image details at different spatial frequencies, and it is an important parameter for evaluating the quality of remote-sensing images. According to ISO 12233 [21], the tilted edge method is a standard method for testing the resolution of electronic still image cameras [22], in which the response of an edge (usually a black and white demarcation line) is analyzed by placing it at an angle in the field of view of the imaging system, using gradient analysis.

Since the experimental images achieved the highest combined objective evaluation index at 24 integration stages, we analyzed the MTF of the TDI images obtained from both the uncompensated method and our algorithm. Figure 15 and Figure 16 illustrate the MTF in the across and along-track directions, measured using the tilted edge method, while Table 7 presents the calculated values for these directions.

It can be seen that the sharpness degradation of the image in the across-track direction is more drastic in the case of no compensation. The MTF of the Baotou region image is 0.08 times the ideal image in the across-track direction and 0.19 times in the along-track direction, and the MTF of the across-track and along-track direction is 0.68 times the ideal image after compensation by our algorithm. The MTF of the Songshan region image in the across-track direction is 0.15 times the ideal image, and the MTF in the along-track direction is 0.21 times the ideal, while the MTF in the across-track and along-track directions after compensation by our algorithm is 0.72 times and 0.69 times the ideal image, respectively. It is proved that the present algorithm can reduce the effects of image motion and jitter on the sharpness of TDI images in both across-track and along-track directions.

In the multispectral TDI imaging experiment, we extended the proposed digital domain TDI algorithm to multispectral remote-sensing images to evaluate its effectiveness with multispectral data. We selected a set of multispectral remote-sensing images from Google Maps containing scenes with various feature types. During the TDI imaging process, we employed the same algorithmic framework as for the panchromatic images, updating the satellite platform and image detector parameters at the time of imaging. Figure 17 shows that the multispectral images obtained using this algorithm are visually superior to uncompensated images. The quantitative analysis in Table 8 reveals that the SSIM of the images produced by this method improved by an average of 0.2886, and the PSNR value increased by an average of 15.02 dB compared to images without compensation, indicating a significant enhancement in imaging quality. These results demonstrate that the proposed algorithm is well-suited and effective for multispectral remote-sensing images.

## 5. Discussion

Firstly, a complete set of digital domain TDI algorithms and processes are proposed for the on-star image-motion model and the jitter mechanism. Compared with the traditional image-motion compensation method, the algorithm (off-chip digital domain self-aligned TDI-CMOS camera) has the advantages of simple optical and mechanical structures and simultaneous elimination of image motion and tremor during the imaging process. The effectiveness of the algorithm is verified by simulation and imaging experiments. Secondly, the experimental results show that compared with the global search, this method can improve the alignment speed and alignment accuracy of inter-frame dark and weak images, and the error of inter-frame global image motion estimation is less than 0.01 pixels, which overcomes the difficulties of the TDI process, such as the large number of calculations of dark and weak image alignment and the large number of false matching points. The simulation and experimental results show that this method can improve the alignment speed of the dark and weak images between frames, and the error of the global image motion estimation between frames is less than 0.01 pixels. Compared with TDI imaging without image motion and jitter compensation, this method has obvious improvement in the indexes of structural similarity, peak signal-to-noise ratio, number of interrelationships, information entropy, and modulation transfer function.

In the future, this algorithm is expected to be embedded in the off-chip digital domain self-aligning TDI-CMOS camera, using the satellite attitude and orbit information and the tremor parameter to achieve the establishment of the minimum search domain in the alignment process, and then to achieve the accurate estimation of the global image motion estimate between frames, in order to release the digital compensation in the TDI imaging process with the help of the on-board processing system.

However, there are still some areas that can be improved: firstly, for the feature point extraction part, this method only uses the existing common operators, which can be optimized for new lightweight operators to adapt to the application scenarios of low illumination and complex texture, as well as to the application requirements of inter-frame displacement alignment. Then, this method can only achieve the motion compensation caused by image motion and jitter in the TDI process, and the experiment uses an ideal image without aberration; however, in actual space-based remote sensing, due to the non-ideal nature of the optical system, there are many kinds of aberrations which lead to distortion of the original image captured by the image detector, yet it can then be combined with the wavefront aberration compensation technology of the optical system and the aberration correction technology of the digital image processing so that it can also achieve aberration compensation during the TDI process.

## 6. Conclusions

The minimum search domain alignment method proposed in this study solves the three core problems of image motion and jitter compensation in TDI imaging by combining the strategy of optimized feature alignment and inter-frame displacement estimation:Joint compensation of image motion and jitter: Optimizing the search domain based on the image motion compensation model and the satellite platform jitter model can eliminate the effects of image motion and jitter on the imaging quality in the imaging process at the same time.Matching feature pairs between frames in the imaging process: Based on the constraints of image motion and jitter, the minimum search domain is used to take the local optimal pair as the global optimal pair, which significantly reduces the time-consumption of the matching process compared with the worldwide search.High-precision global displacement estimation: A kernel density estimation algorithm is proposed to deal with the probability distribution of inter-frame displacements efficiently, and the weighted assignment of each feature pair achieves sub-pixel level displacement estimation with an error of less than 0.01 pixels.

This method achieves inter-frame image alignment in the digital domain, avoids the complexity of traditional external mechanical compensation, and provides an efficient and low-cost solution for TDI-CMOS cameras in aerospace remote sensing. Future research can further combine the back-illumination technique and digital-to-analog conversion method of TDI-CMOS to expand the usage scenarios of non-push-scan imaging modes, and further explore methods such as deep learning to optimize sub-pixel-level self-compensation in the digital domain.

## Figures and Tables

**Figure 1 sensors-25-03490-f001:**
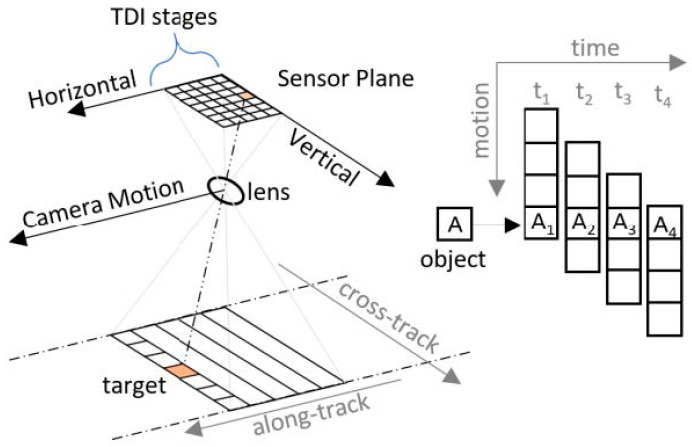
Schematic diagram of space-based TDI imaging.

**Figure 2 sensors-25-03490-f002:**
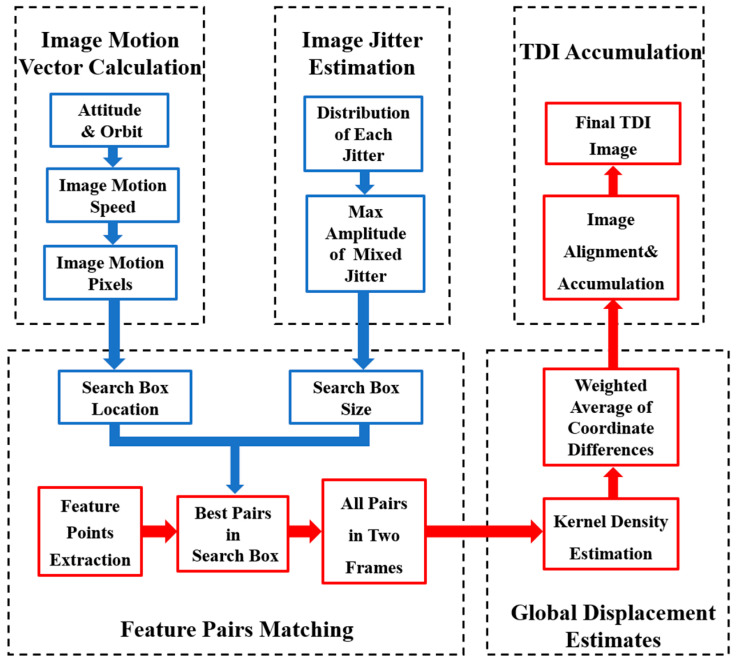
Block diagram of TDI imaging method flow based on minimum search domain.

**Figure 3 sensors-25-03490-f003:**
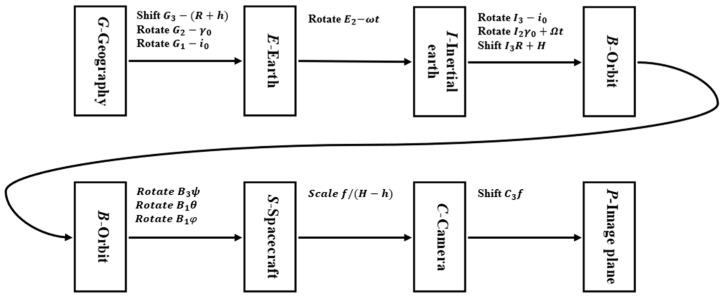
Transformation relationships between major coordinate systems in Earth imaging.

**Figure 4 sensors-25-03490-f004:**
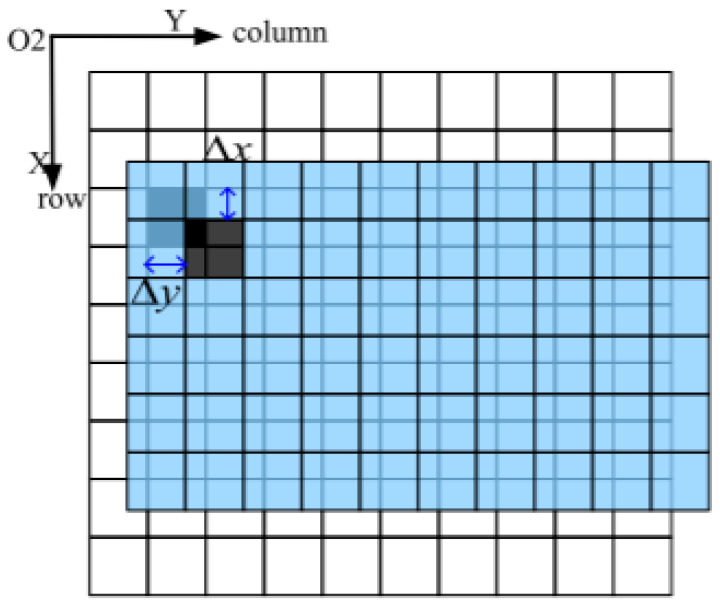
Diagram of image motion at the focal plane.

**Figure 5 sensors-25-03490-f005:**
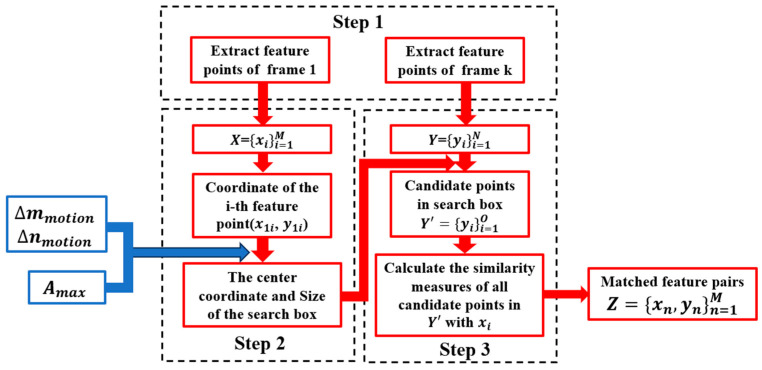
Flowchart of feature point pair matching.

**Figure 6 sensors-25-03490-f006:**
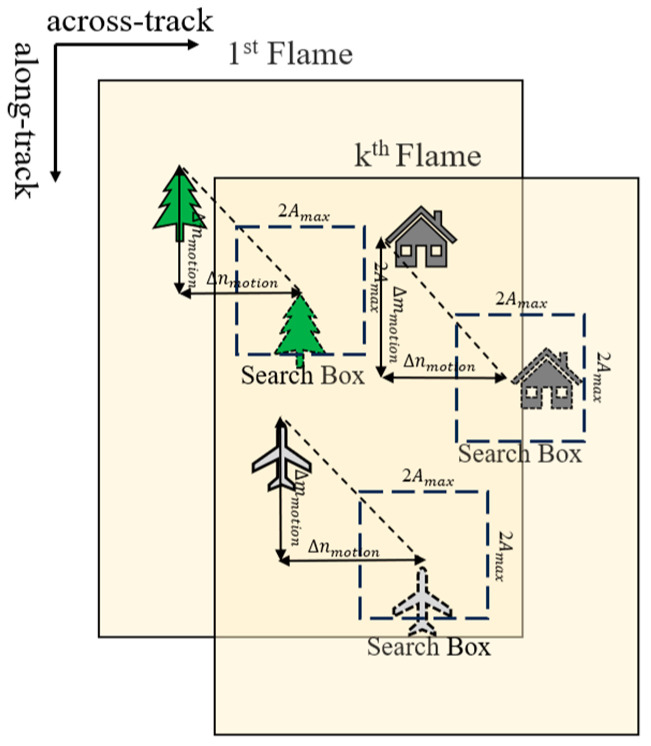
Schematic of the search box.

**Figure 7 sensors-25-03490-f007:**
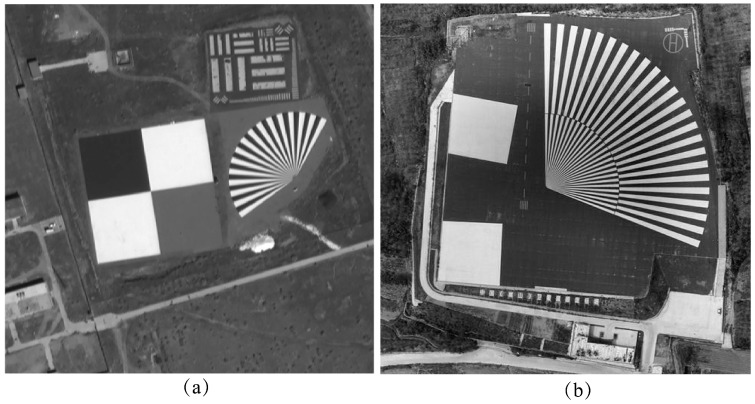
Raw remote sensing imagery of the target range. (**a**) Baotou range; (**b**) Songshan range.

**Figure 8 sensors-25-03490-f008:**
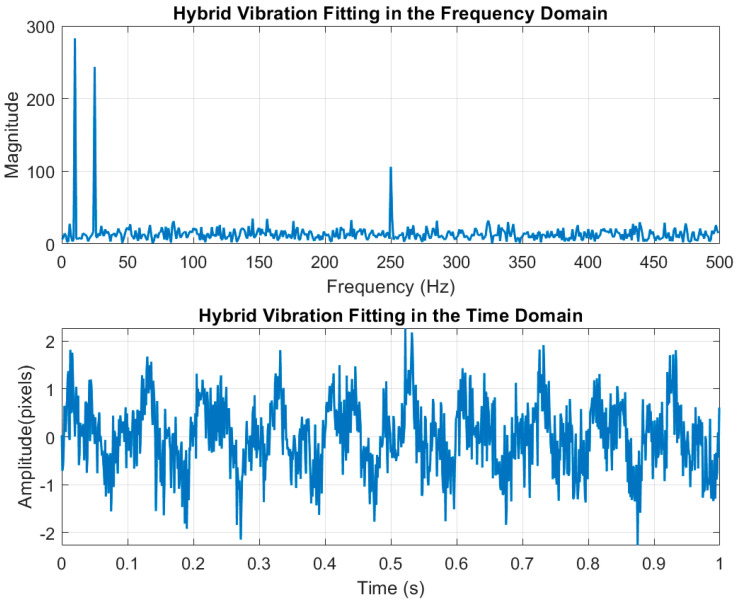
Frequency-domain and time-domain fitting plots for hybrid jitters.

**Figure 9 sensors-25-03490-f009:**
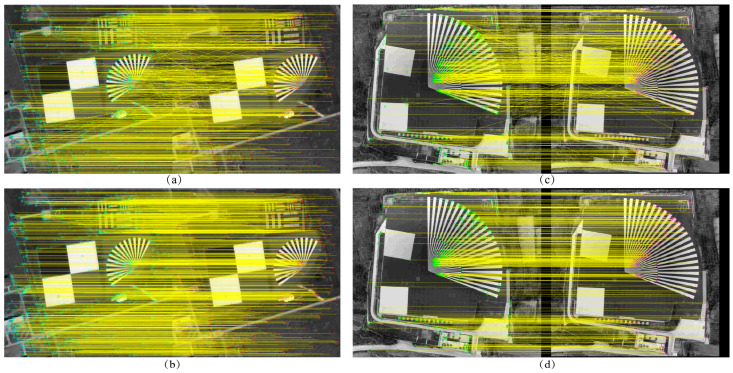
Comparison of the effect of inter-frame image alignment (**a**) Global matching of Baotou range (**b**) Minimum search domain matching of Baotou range (**c**) Global matching of Songshan range (**d**) Minimum search domain matching of Songshan range.

**Figure 10 sensors-25-03490-f010:**
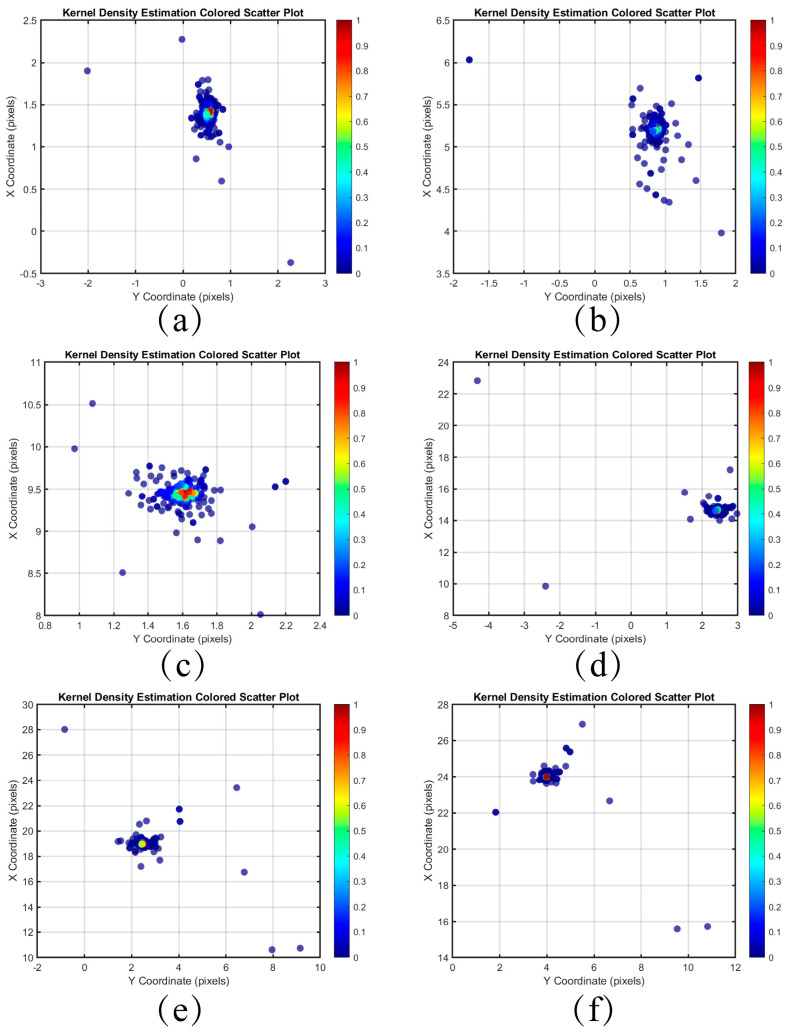
Kernel density estimation distribution of difference in coordinate pairs of feature points between each frame and the first frame using SIFT only. (**a**) Frame 2 (**b**) Frame 6 (**c**) Frame 10 (**d**) Frame 15 (**e**) Frame 20 (**f**) Frame 24.

**Figure 11 sensors-25-03490-f011:**
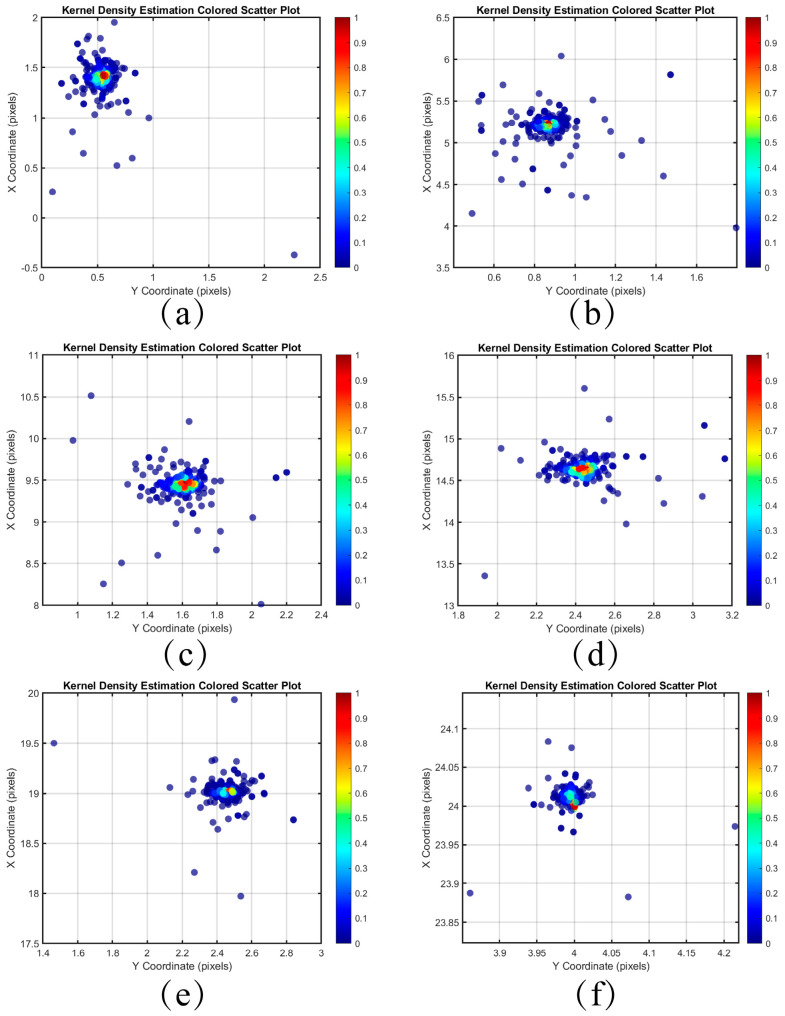
Kernel density estimation distribution of coordinate differences between each frame and the first frame feature point pair for the minimum search domain. (**a**) Frame 2 (**b**) Frame 6 (**c**) Frame 10 (**d**) Frame 15 (**e**) Frame 20 (**f**) Frame 24.

**Figure 12 sensors-25-03490-f012:**
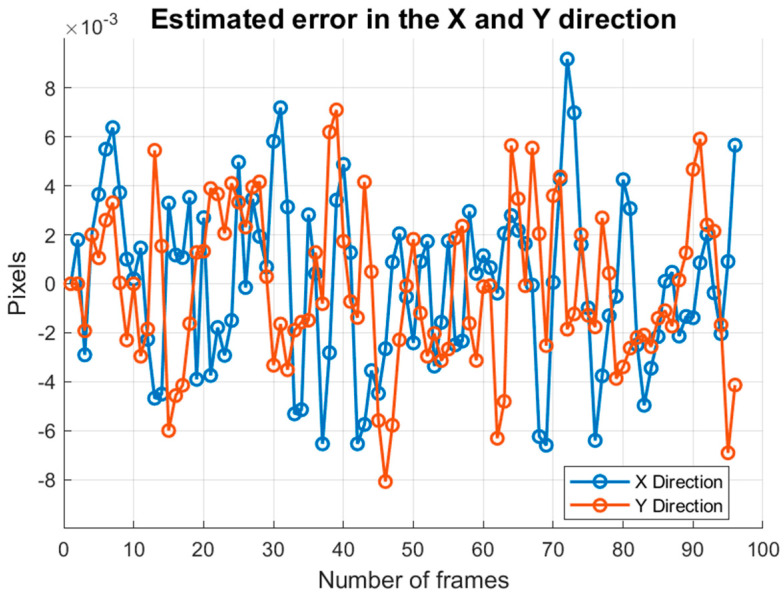
Error analysis of the estimated displacement values for each frame.

**Figure 13 sensors-25-03490-f013:**
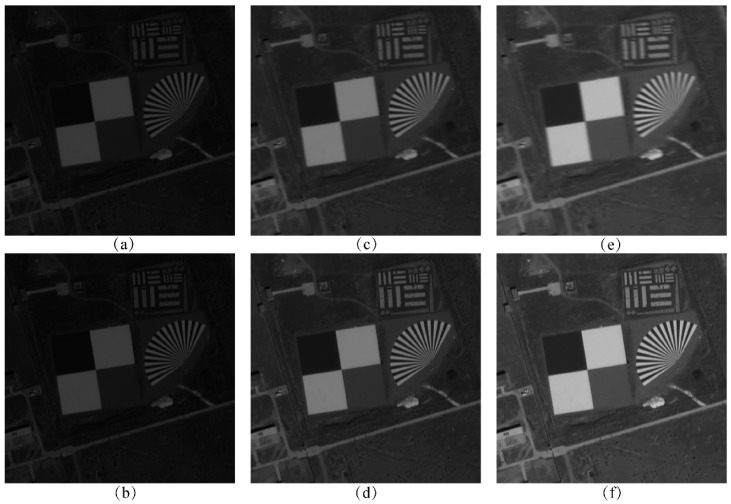
TDI stacking effects for 6–18 stages: (**a**) 6-stage without compensation; (**b**) 6-stage with our algorithm; (**c**) 12-stage without compensation; (**d**) 12-stage with our algorithm; (**e**) 18-stage without compensation; (**f**) 18-stage with our algorithm.

**Figure 14 sensors-25-03490-f014:**
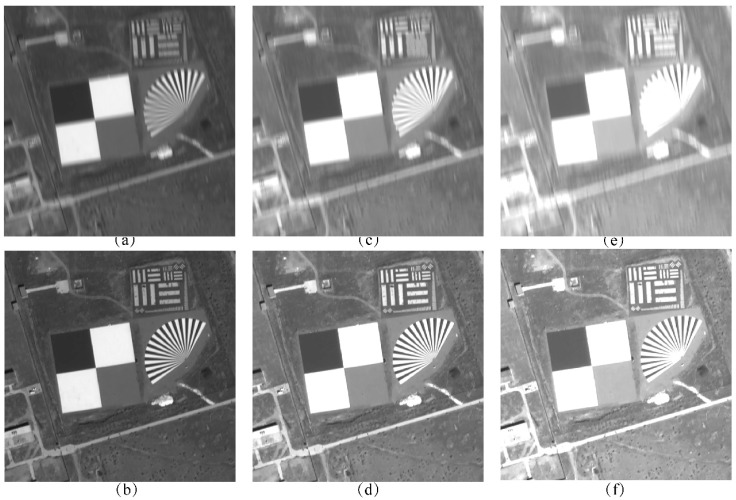
TDI stacking effects for 24–48 stages: (**a**) 24-stage without compensation; (**b**) 24-stage with our algorithm; (**c**) 36-stage without compensation; (**d**) 36-stage with our algorithm; (**e**) 48-stage without compensation; (**f**) 48-stage with our algorithm.

**Figure 15 sensors-25-03490-f015:**
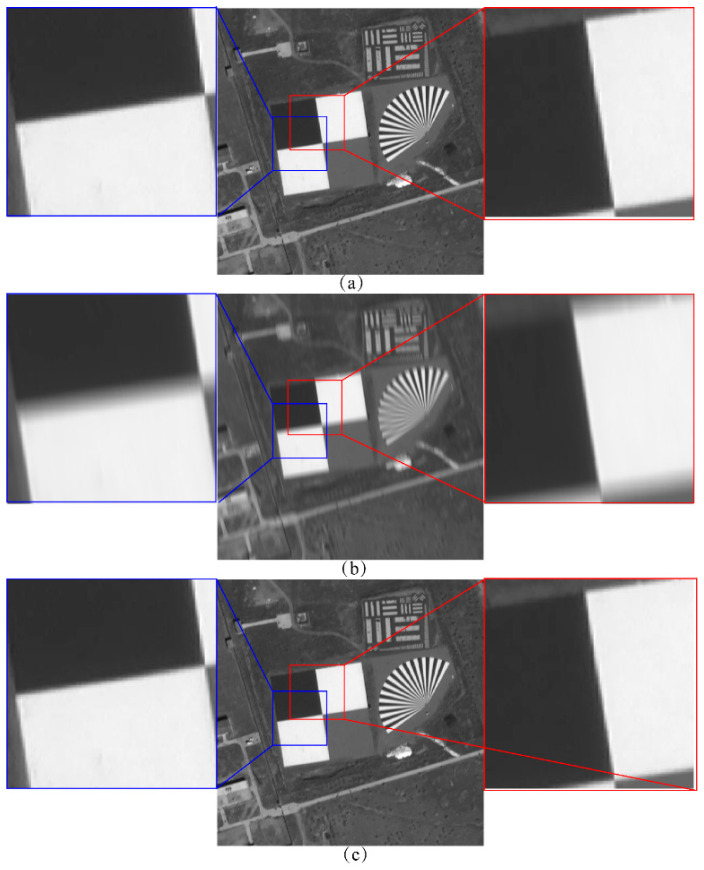
MTF of Baotou region images measured by the tilted edge method: (**a**) Ideal image; (**b**) The image without image motion and jitter compensation; (**c**) The image obtained by our algorithm.

**Figure 16 sensors-25-03490-f016:**
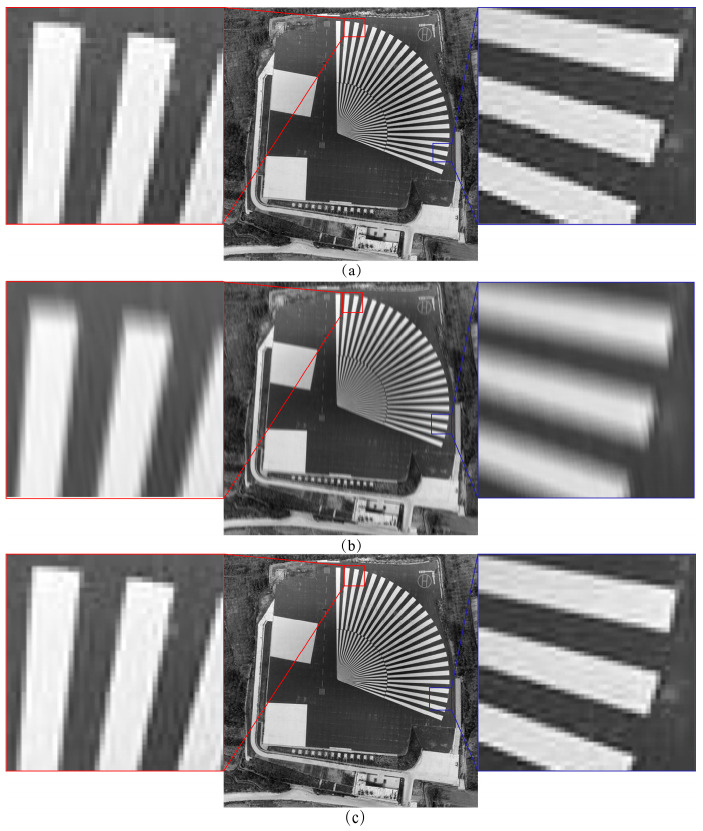
MTF of Songshan region images measured by the tilted edge method: (**a**) Ideal image; (**b**) The image without image motion and jitter compensation; (**c**) The image obtained by our algorithm.

**Figure 17 sensors-25-03490-f017:**
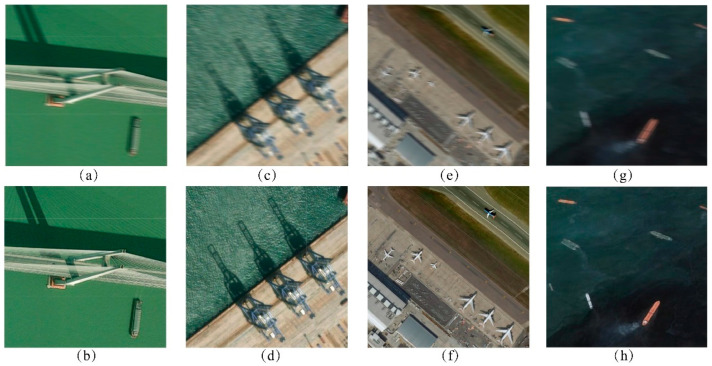
Multi-spectral TDI imaging results: (**a**) Bridge without compensation; (**b**) Bridge with our algorithm; (**c**) Harbor without compensation; (**d**) Harbor with our algorithm; (**e**) Airport without compensation; (**f**) Airport with our algorithm; (**g**) Ships without compensation; (**h**) Ships with our algorithm.

**Table 1 sensors-25-03490-t001:** Platform jitter parameter types.

Platform Jitter	Distribution	Parameter
Low-frequency	Sine function	$f_{Low\text{-}frequency},A_{Low\text{-}frequency}$
High-frequency	Sine function	$f_{High\text{-}frequency},A_{High\text{-}frequencyy}$
Unexpected	Normal distribution	μ,σ

**Table 2 sensors-25-03490-t002:** Spacecraft and image detector indicators.

Symbol	Spacecraft Parameters	Measurements
f	Focal length	0.8888 m
H	The altitude of the spacecraft to the nadir point	$5.2997\times 10^{5}$ m
h	Terrain Height at the nadir point	1064.91 m
φ	Roll angle	0.05047 rad
θ	Pitch angle	0.06615 rad
ψ	Yaw angle	−0.1622 rad
$\gamma_{0}$	Ascension of ascending node	0.8307 rad
Ω	Orbital angular velocity of the spacecraft	$1.1071\times 10^{-3}$ rad/s
$i_{0}$	Orbital inclination	1.7038 rad
a	Detector pixel size	$3.2\times 10^{-6}\;m$/pixel
$T_{stage}$	Exposure time	$2.6752\times 10^{-4}$ s

**Table 3 sensors-25-03490-t003:** Satellite platform jitter type and parameter indicators.

Platform Jitter	Distribution	Sample Parameter
Low-frequency	Sine function	$f_{1}=10\;Hz,\;A_{1}=0.6\;pixel$
		$f_{2}=25\;Hz,\;A_{2}=0.5\;pexel$
High-frequency	Sine function	$f_{3}=250\;Hz,\;A_{3}=0.2\;pixel$
		$f_{4}=500\;Hz,\;A_{4}=0.1\;pixel$
Unexpected	Normal distribution	μ=0 pixel, σ=0.2 pixel

**Table 4 sensors-25-03490-t004:** Feature point pair matching and durations.

Number of Feature Points	Equivalent Global SIFT Time-Consumed (ms)	Search Box Size (pixel × pixel)	Our Time- Consumed (ms)
1256	132.24	4 × 4	**81.76**
		8 × 8	116.55
		10 × 10	122.06
4891	325.97	4 × 4	101.06
		8 × 8	170.74
		10 × 10	256.68
25461	684.11	4 × 4	**177.32**
		8 × 8	278.71
		10 × 10	350.75

**Table 5 sensors-25-03490-t005:** Analysis of global displacement estimates for each frame with respect to the first frame.

	Frame 2	Frame 6	Frame 10	Frame 15	Frame 20	Frame 24
True displacement (pixel)	(1.4093, 0.5384)	(5.2223, 0.8678)	(9.4540, 1.6159)	(14.6345, 2.4418)	(19.0130, 2.4657)	(24.0169, 3.9860)
Equivalent Global SIFT Estimation						
Estimated displacement (pixel)	(1.4073, 0.5383)	(5.2166, 0.8658)	(9.4548, 1.6161)	(14.6302, 2.4381)	(19.0158, 2.4594)	(24.0056, 3.9915)
Error value (pixel)	(0.0020, 0.0001)	(0.0057, 0.0020)	(−0.0008, −0.0002)	(0.0043, 0.0037)	(−0.0028, 0.0063)	(0.0113, −0.0055)
Our Estimation						
Estimated displacement (pixel)	(1.4075, 0.5384)	(5.2168, 0.8652)	(9.4544, 1.6163)	(14.6323, 2.4385)	(19.0142, 2.4630)	(24.0104, 3.9875)
Error value (pixel)	(0.0018, 0.0000)	(0.0055, 0.0026)	(−0.0004, −0.0004)	(0.0022, 0.0033)	(−0.0012, 0.0027)	(0.0065, −0.0015)

**Table 6 sensors-25-03490-t006:** Indicators for objective evaluation of TDI imaging results.

Integral Stage	Without Compensation				Ours			
	SSIM	PSNR	CC	Entropy	SSIM	PSNR	CC	Entropy
6	0.3963	10.0326	0.9459	4.4732	0.4200	10.0817	0.9999	4.4836
12	0.6640	13.2440	0.9042	5.4234	0.7638	13.6020	1.0000	5.4523
18	0.7754	17.5582	0.8765	5.9978	0.9495	19.6184	1.0000	6.0190
24	0.7886	20.5139	0.8583	6.3936	0.9985	35.4656	1.0000	6.4002
36	0.6997	12.6242	0.8249	6.4860	0.9028	13.6095	1.0000	6.6033
48	0.5871	7.6784	0.7823	6.1012	0.7572	7.9569	1.0000	6.3990

**Table 7 sensors-25-03490-t007:** Across-track and along-track MTF analysis.

Imagery	Types of Imagery	Across-Track MTF (Ratios)	Along-Track MTF (Ratios)
Baotou region	Ideal	0.02552 (1.0000)	0.03941 (1.0000)
	Without Compensation	0.002089 (0.08190)	0.007603 (0.1934)
	Our	0.01731 (0.6783)	0.02684 (0.6810)
Songshan region	Ideal	0.02978 (1.0000)	0.01187 (1.0000)
	Without Compensation	0.004467 (0.1500)	0.002480 (0.2089)
	Our	0.02133 (0.7163)	0.08189 (0.6899)

**Table 8 sensors-25-03490-t008:** Indicators for objective evaluation of multispectral imaging results.

Types of Images	Without Compensation				Ours			
	SSIM	PSNR	CC	Entropy	SSIM	PSNR	CC	Entropy
Bridge	0.7845	22.8762	0.8195	5.4478	0.9710	37.0480	0.9936	5.7456
Harbor	0.5127	20.8526	0.8696	7.1541	0.9681	36.6603	0.9967	7.3754
Airport	0.6268	21.0712	0.8382	6.8760	0.9410	32.8660	0.9900	7.0440
Ships	0.7838	20.4969	0.8535	6.2193	0.9821	38.8098	0.9980	6.3710

## Data Availability

The original contributions presented in this study are included in the article. Further inquiries can be directed to the corresponding author.

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
