# Peer review of "Digital Domain TDI-CMOS Imaging Based on Minimum Search Domain Alignment"

_sensors, 2025, doi:10.3390/s25113490_

Round 1
Reviewer 1 Report
Comments and Suggestions for Authors
Abstract is a detailed description of the proposed method and brief conclusions based on the results of the article. The abstract should be a total of about 200 words maximum. Place the question addressed in a broad context and highlight the purpose of the study.
Introduction is a brief overview of modern methods of image motion compensation, as well as current primary methods exist to compensate for the effects of satellite plat form jitter. Several references to articles in highly ranked journals with a publication date after 2020 are provided. The main objective of the work is highlighted and the main conclusions are highlighted.
In the Digital Image Domain TDI-CMOS Operating Principle section presents the working principle of the proposed methods, which the authors propose to use for imaging technology in remote sensing.
Point 1. Fig. 1. Not specified in the text, reference to Fig. 1 and no explanations of the Schematic diagram of space-based TDI imaging.
In the TDI Imaging Method Based on Minimum Search Domain section, each imaging process of this method is described.
Point 2. Fig. 3–5. Not specified in the text, references to Fig. 3–5 and no explanation of the parameters indicated on them. It is necessary to add a description of the variables G, E, I, B, C, their values, physical meaning.
Point 3. Eq.1,2. It is necessary to add a description of the parameters P, P1…P3.
Point 4. Eq.3–6. It is necessary to add a description of the parameter t0.
Point 5. Eq.5,6. It is necessary to explain the reason for preserving the integral sum when moving to the discrete case. Perhaps it is necessary to replace the integral sum with a series. The parameter “a” is not described.
Point 6. Eq.16,17. The operator “fix” is not defined. Does it remove the fractional part of the number and return the resulting integer value? It must be specified explicitly.
The Experiments section provides a simulation experiment of the digital domain TDI-CMOS 96 imaging process. It is worth noting that Table 2 presents Spacecraft and image detector indicators, which allow us to talk about the reproducibility of the results.
Point 7. Fig. 9, 10. Unreadable captions on figures. Needs to be redone.
Point 8. Fig. 12. Not specified in the text of references to Fig. 12.
Point 9. Tab. 6. It is necessary to describe the Indicators for objective evaluation (SSIM, PSNR, CC, entropy) or provide Eq. for calculating these indicators or provide refs to primary sources.
Point 10. In the Discussion section, it is necessary to add the disadvantages of the presented method. It is proposed to conduct a comparison with existing technologies based on compensation of wavefront aberrations. You could describe further prospects for the development of the proposed method.
In the Conclusions section, the authors indicated the main conclusions. The main result is the compensation of low-frequency and random oscillations that simulated the shaking of the satellite platform during image acquisition. The article demonstrates that the present algorithm effectively reduces the impact of motion and image tremor on image clarity.
Despite the indicated shortcomings, I believe that this article can be published after a minor revision.
Author Response
Comments 1: Fig. 1. Not specified in the text, reference to Fig. 1 and no explanations of the Schematic diagram of space-based TDI imaging. |
Response 1: Thank you for pointing this out and we agree with this comment. We have added a reference to Figure 1 and an interpretation of the starboard TDI imaging schematic to the paper. The specific additions are on page 3, lines 116-120. |
Comments 2: Fig. 3–5. Not specified in the text, references to Fig. 3–5 and no explanation of the parameters indicated on them. It is necessary to add a description of the variables G, E, I, B, C, their values, physical meaning. Response 2: Agreeing with your comment, for Figures 3 - 5, we have added a description of them on page 5, lines 148-155, with a detailed explanation of the coordinate systems G, E, I, B, and C in the figures. In the added paragraph, the physical meaning of each coordinate system and the process of its transformation is elaborated to help the reader better understand it. |
Comments 3: Eq.1,2. It is necessary to add a description of the parameters P, P1…P3. |
Response 3: Agree with your comment adding the description of P, P1...P3 in Eq.1,2. as the coordinate values specific to the like plane coordinate system. |
Comments 4: It is necessary to add a description of the parameter t0 |
Response 4: This problem is solved with comment 5, where the initial photo moment t0 does not exist after changing the Eq. 3- Eq. 6 integrals to levels. |
Comments 5: Eq.5,6. It is necessary to explain the reason for preserving the integral sum when moving to the discrete case. Perhaps it is necessary to replace the integral sum with a series. The parameter “a” is not described. |
Response 5: Agree with your comment that Eq.3- Eq.6 have been replaced with expressions in the form of levels, while the physical meaning of the parameter ‘a’ is the detector element size. |
Comments 6: The operator “fix” is not defined. Does it remove the fractional part of the number and return the resulting integer value? It must be specified explicitly. |
Response 6: Agree with your comment that Eq.16- Eq.17 have been replaced with mathematically explicit expressions. The fix function, which previously used matlab rounding, does indeed mean rounding. |
Comments 7: Fig. 9, 10. Unreadable captions on figures. Needs to be redone. Response 7: We have reformatted figures 9 and 10 to ensure that the captions are readable. |
Comments 8: Fig. 12. Not specified in the text of references to Fig. 12. Response 8: A description of Fig. 12. has been added, specifically at lines 357-362 on page 15. |
Comments 9: It is necessary to describe the Indicators for objective evaluation (SSIM, PSNR, CC, entropy) or provide Eq. for calculating these indicators or provide refs to primary sources. Response 9: Above Table 6, we have added a detailed description of the objective evaluation metrics (SSIM, PSNR, CC, entropy), as specified on page 16, rows 374-383. The reference source [20] is also labelled to facilitate the reader's further access to the relevant formulas. |
Comments 10: In the Discussion section, it is necessary to add the disadvantages of the presented method. It is proposed to conduct a comparison with existing technologies based on compensation of wavefront aberrations. You could describe further prospects for the development of the proposed method. Response 10: Agreeing with your comment, in the ‘Discussion’ section, we have added a new paragraph, specifically on page 20, lines 468-480. It is devoted to the shortcomings of the proposed method, including the study of better feature point extraction operators and aberration compensation, and the joint of wavefront aberration compensation is discussed. We also extend the development prospects and applications of the proposed method.
|
Reviewer 2 Report
Comments and Suggestions for Authors
This paper presents an innovative digital TDI-CMOS imaging method, specifically designed to mitigate the effects of image motion and satellite platform instability on TDI (Time Delay Integration) images. The method introduced in this study utilizes an alignment strategy centered on a restricted search domain, making it highly compatible with the area-array push-scan imaging configuration. This technique enhances the fusion of low-contrast and dark images by refining the alignment procedure and accurately determining the global frame-to-frame displacement. A notable advantage of the approach is its ability to compensate for both image motion and platform jitter concurrently, without the need for mechanical systems to adjust angular deviations. The entire process is structured into five implementation steps, which are described in detail throughout the article.
The paper is well written, clearly, and well organized. The scientific level is high and well sustained by a good mathematical support, convincing results, like high precision displacement estimation, and it may be accepted after the minor corrections suggested to the authors.
- The abbreviations MTF (Modulation Transfer Function) and SIFT (Scale-Invariant Feature Transform) should be clearly explained at their first occurrence in the text to ensure clarity for all readers.
- Several figures are included in the manuscript without being explicitly referenced or discussed in the text. Examples include Figure 1, Figure 2, Figure 3, Figure 4, Figure 5, Figure 6, Figure 11, and Figure 12. Each figure should be clearly introduced and interpreted within the main body of the text.
- There is also a lack of textual reference and explanation for certain tables, such as Table 7, which should be properly contextualized and discussed in the corresponding section.
- In some cases, the explanation in the text does not match the content of the subsequent figures, leading to inconsistencies in interpretation. This should be carefully reviewed and corrected to ensure alignment between the narrative and the visual elements.
- The content between lines 204 and 208 lacks clarity and should be reformulated to improve readability and comprehension.
- Section 3.1 begins directly with Figure 3, without a prior mention or contextual introduction of the figure in the narrative. It is recommended to introduce and describe each figure before it appears.
- In Section 3.4, the title is misspelled: "Global Dispalcement Estimate" should be corrected to "Global Displacement Estimate."
- With reference to Figure 14, the explanation provided regarding the MTF is insufficient. It is suggested to include an introductory paragraph earlier in the manuscript explaining the tilted edge method, which is commonly used for MTF measurement (e.g., according to ISO 12233 standards).
- Regarding Equation (8), it would be helpful to cite relevant bibliographic sources that support or previously employed this approach, to strengthen the theoretical foundation of the method
Author Response
Comments 1: The abbreviations MTF (Modulation Transfer Function) and SIFT (Scale-Invariant Feature Transform) should be clearly explained at their first occurrence in the text to ensure clarity for all readers. |
Response 1: Thank you for pointing this out and we agree with this comment. We have added full explanations of MTF (Modulation Transfer Function) and SIFT (Scale - Invariant Feature Transform) in the paper where they first appeared, to ensure that all readers can quickly understand their meanings and to enhance the readability of the paper. The readability of the paper is enhanced by adding complete explanations in English and Chinese to ensure that all readers can quickly understand the meaning. Specifically on page 18, lines 398-401 and page 7, lines 210-211. |
Comments 2: Several figures are included in the manuscript without being explicitly referenced or discussed in the text. Examples include Figure 1, Figure 2, Figure 3, Figure 4, Figure 5, Figure 6, Figure 11, and Figure 12. Each figure should be clearly introduced and interpreted within the main body of the text. Response 2: Agree with your comment and have made changes, we have added clear references and descriptions to the icons in the text, e.g. the description of Figure 1 is on page 3, lines 116-120, the description of Figure 2 is on page 4, lines 132-135, the description of Figure 3 is on page 5, lines 149-155 etc. |
Comments 3: There is also a lack of textual reference and explanation for certain tables, such as Table 7, which should be properly contextualized and discussed in the corresponding section. |
Response 3: Changes have been made, and for Table 7 and other similar tables, we have also added the characteristics and significance of the data in the table, specifically on page 18, lines 411-421. |
Comments 4: In some cases, the explanation in the text does not match the content of the subsequent figures, leading to inconsistencies in interpretation. This should be carefully reviewed and corrected to ensure alignment between the narrative and the visual elements. |
Response 4: Regarding the consistency of text and diagrams: We reviewed the paper sentence by sentence and diagram by diagram, and carefully checked and corrected any inconsistencies in interpretation. We have reorganised the textual narrative and the content of the figures to ensure that they echo each other and are logically coherent, so as to avoid causing misunderstanding to the readers. For example, for Figure 6, the text description was optimised on page 7, lines 214-219, to make the two more closely related. |
Comments 5: The content between lines 204 and 208 lacks clarity and should be reformulated to improve readability and comprehension. |
Response 5: Agree with your comment. We have reformulated the content of lines 204 - 208 by simplifying complex sentences, splitting lengthy paragraphs, and adding the necessary transitional statements and explanatory notes to make the content of this section clearer and more logical. The changed paragraph is split into four paragraphs on page 7, lines 207-223. |
Comments 6: Section 3.1 begins directly with Figure 3, without a prior mention or contextual introduction of the figure in the narrative. It is recommended to introduce and describe each figure before it appears |
Response 6: In Section 3.1, we have added a paragraph describing the principle and process of coordinate system transformation proposed in the literature13 before referring to Fig. 3, specifically on page 5, lines 148-155, to facilitate a combined graphical understanding. |
Comments 7: In Section 3.4, the title is misspelled: "Global Dispalcement Estimates" should be corrected to "Global Displacement Estimate." Response 7: Changes have been made according to your request. |
Comments 8: Fig. 12. With reference to Figure 14, the explanation provided regarding the MTF is insufficient. It is suggested to include an introductory paragraph earlier in the manuscript explaining the tilted edge method, which is commonly used for MTF measurement (e.g., according to ISO 12233 standards). Response 8: We have added a new introductory paragraph on MTF on page 18, lines 398-405 of the paper, which describes the principles and measurements of MTF and the necessity of using it for image evaluation, as well as the standards associated with the measurement method. |
Comments 9: Regarding Equation (8), it would be helpful to cite relevant bibliographic sources that support or previously employed this approach, to strengthen the theoretical foundation of the method Response 9: Thanks to your suggestion, we have added references 10 and 14, which were used to give convincing arguments for the type of satellite platform tremor and tremor parameters for the simulation experiment setup. |
Reviewer 3 Report
Comments and Suggestions for Authors
This manuscript presents a digital-domain TDI-CMOS dynamic imaging method based on minimum search domain alignment, aiming to address the issue of the influence of image motion and platform jitter on TDI imaging quality in satellite remote sensing imaging. The research content has certain application value and the experimental design is relatively reasonable. However, I believe that the digital-domain TDI-CMOS imaging method proposed in the manuscript based on minimum search domain alignment is not innovative enough. It merely combines existing image motion compensation models, satellite platform jitter models, common feature point matching algorithms such as SIFT, and kernel density estimation algorithms without proposing unique and novel ideas or key improvements. Therefore, this paper is not suitable for publication in "SENSORS". The detailed comments are as follows.
- The abstract is logically confused and fails to describe clearly the problem that this manuscript intends to address.
- In the Introduction section, the connections between different parts are not tight enough and the logical relationship is not clear. For instance, from the introduction of the significance of high-resolution optical remote sensing satellites and the application of TDI-CMOS devices, to the elaboration of the influence of image motion and satellite platform jitter on imaging quality, and then to the shortcomings of existing compensation methods and the proposal of the current method, the transitions between the parts are not smooth. Moreover, when introducing the existing methods for image motion compensation and satellite platform jitter compensation, only various methods and their problems are simply listed without an in-depth analysis and comparison of the intrinsic principles, advantages and disadvantages of these methods.
- When elaborating on the digital domain TDI-CMOS imaging technology proposed in the manuscript, it merely indicates that this technology solves the problem by using minimum search domain alignment, but fails to highlight the specific manifestations of its innovation and unique advantages compared with existing methods. It does not clearly explain how the method based on minimum search domain alignment can more effectively overcome the shortcomings of existing methods in optimizing the image alignment process and estimating inter-frame global displacements.
- In the calculation of image motion vectors, although complex calculation formulas have been provided, the physical meanings and interrelationships of each parameter in the formulas are not explained clearly enough.
- In the step of feature point pair matching, the SIFT algorithm is selected to extract feature points, but its limitations in complex environments are not fully considered. The paper only mentions the advantages of the SIFT algorithm without analyzing the problems that may occur in actual imaging under low illumination and complex backgrounds, such as incomplete feature point extraction and increased mis-matching.
- In the estimation of global displacement, the Kernel Density Estimation (KDE) algorithm is employed to calculate the probability density, but the method for selecting the key parameters in the KDE algorithm is not explained.
- The experiment only selected satellite remote sensing images from Baotou region, and the sample was too monotonous. The types of ground objects, texture features and lighting conditions vary greatly among different regions. Images from a single region cannot fully verify the performance of the algorithm in diverse environments.
Author Response
1. Summary |
|
|
Thank you for reviewing my thesis in such a detailed and professional manner and offering valuable suggestions for revision, which are very detailed and to the point, and point out a clear direction for us to further improve the thesis and subsequent in-depth research. However, regarding your comment that this thesis does not propose unique and novel ideas or key improvements, I believe that the integration of existing methods is in itself a kind of innovation, especially the systematic innovation in specific application scenarios. Just as for remote sensing satellites, whose attitude is not related to the orbit data and inter-frame images per se, the method in this paper was able to use the attitude-orbit data to greatly reduce the search area to accommodate the microsecond computational response requirements between two frames of the TDI process, which achieves the desired goal. Moreover, this article is the first small paper I wrote as a graduate student, what I want to do is to propose a framework first, in the subsequent research can continue to improve this framework and solve other application scenarios may encounter problems, such as non-substellar point imaging dynamic imaging mode, and non-global shutter exposure method of the roll-up shutter, etc., and of course, including the problem of the feature extraction operator that you mentioned. It would be difficult to cram all of these into one MDPI sensor in terms of space, so I hope you can understand part of my work so far, and welcome your guidance on my subsequent research work! Here is my reply: |
||
2. Point-by-point response to Comments and Suggestions for Authors |
||
Comments 1: The abstract is logically confused and fails to describe clearly the problem that this manuscript intends to address. |
||
Response 1: Thank you for pointing this out and we agree with this comment. We have reorganized the structure and content of the abstract, further condensed it to meet the journal's requirement of around 200 words, summarized the problems addressed throughout the imaging process, and further condensed the conclusions obtained from this study. The specific changes are on the first page, lines 13-26. |
||
Comments 2: In the Introduction section, the connections between different parts are not tight enough and the logical relationship is not clear. For instance, from the introduction of the significance of high-resolution optical remote sensing satellites and the application of TDI-CMOS devices, to the elaboration of the influence of image motion and satellite platform jitter on imaging quality, and then to the shortcomings of existing compensation methods and the proposal of the current method, the transitions between the parts are not smooth. Moreover, when introducing the existing methods for image motion compensation and satellite platform jitter compensation, only various methods and their problems are simply listed without an in-depth analysis and comparison of the intrinsic principles, advantages and disadvantages of these methods. Response 2: For the introduction part, we have reorganized the language, strengthened the relationship between paragraphs, and introduced the mechanism of image motion and jitter and the existing compensation methods from the two main factors of motion blur, and discussed the problems of a single method, and then led to the advantages of this method, so as to achieve the effect of the introduction. |
||
Comments 3: When elaborating on the digital domain TDI-CMOS imaging technology proposed in the manuscript, it merely indicates that this technology solves the problem by using minimum search domain alignment, but fails to highlight the specific manifestations of its innovation and unique advantages compared with existing methods. It does not clearly explain how the method based on minimum search domain alignment can more effectively overcome the shortcomings of existing methods in optimizing the image alignment process and estimating inter-frame global displacements. |
||
Response 3: For the unique advantages of this method over existing methods, we add the alignment effect experiments in Fig. 9, which, combined with the computational time-consumption analysis in Table 4, highlight the speed and accuracy advantages of the minimal search domain method. The details are in lines 317-324 and 330-332 on page 13. |
||
Comments 4: In the calculation of image motion vectors, although complex calculation formulas have been provided, the physical meanings and interrelationships of each parameter in the formulas are not explained clearly enough. |
||
Response 4: In Section 3.1, we add a paragraph describing the principle and process of the coordinate system transformation proposed in the literature13 before referring to Fig. 3, specifically on page 5, lines 148-155, to facilitate the combination of illustrations and text to understand the complex image-shift calculation formula. |
||
Comments 5: In the step of feature point pair matching, the SIFT algorithm is selected to extract feature points, but its limitations in complex environments are not fully considered. The paper only mentions the advantages of the SIFT algorithm without analyzing the problems that may occur in actual imaging under low illumination and complex backgrounds, such as incomplete feature point extraction and increased mis-matching. |
||
Response 5: I agree with your forward-looking considerations. It is true that the SIFT operator is limited in dark weak environments, and the number of feature points that can be extracted is greatly reduced, so we have supplemented the requirements for setting the parameters of the SIFT operator in this environment, which are described in detail on page 12, 303-307. However, for other problems that may be encountered, such as incomplete feature point extraction and increased mismatches, a new paper can be started separately. |
||
Comments 6: In the estimation of global displacement, the Kernel Density Estimation (KDE) algorithm is employed to calculate the probability density, but the method for selecting the key parameters in the KDE algorithm is not explained. |
||
Response 6: In fact we use the minimum integral mean square error to determine the theoretically optimal bandwidth, the additional description is on page 9, lines 248-251, due to space constraints we cannot expand the description, further in-depth research may be presented in the next paper. |
||
Comments 7: The experiment only selected satellite remote sensing images from Baotou region, and the sample was too monotonous. The types of ground objects, texture features and lighting conditions vary greatly among different regions. Images from a single region cannot fully verify the performance of the algorithm in diverse environments. Response 7: Subsequently, we supplemented the remote sensing images of the Songshan target range, and the final TDI effect of the simulation is shown in Fig. 16, and the imaging index is still close to the Baotou target range, which did not affect the conclusions given. Since remote sensing images in the natural environment where there are straight, black and white, and easy to select the edge region to measure the MTF value are very rare, only target range images can be used in general. |
Round 2
Reviewer 3 Report
Comments and Suggestions for Authors
I have no problem.